# FiCoS: A fine-grained and coarse-grained GPU-powered deterministic simulator for biochemical networks

**Andrea Tangherloni**[1]*, **Marco S. Nobile**[2,3,4], **Paolo Cazzaniga**[1,3,4], **Giulia Capitoli**[4,5], **Simone Spolaor**[4,6], **Leonardo Rundo**[7,8], **Giancarlo Mauri**[3,4,6], **Daniela Besozzi**[3,4,6]*

**1** Department of Human and Social Sciences, University of Bergamo, Bergamo, Italy, **2** Department of Industrial Engineering & Innovation Sciences, Eindhoven University of Technology, Eindhoven, The Netherlands, **3** SYSBIO/ISBE.IT Centre of Systems Biology, Milan, Italy, **4** Bicocca Bioinformatics Biostatistics and Bioimaging Centre (B4), University of Milano-Bicocca, Vedano al Lambro, Italy, **5** School of Medicine and Surgery, University of Milano-Bicocca, Monza, Italy, **6** Department of Informatics, Systems and Communication, University of Milano-Bicocca, Milan, Italy, **7** Department of Radiology, University of Cambridge, Cambridge, United Kingdom, **8** Cancer Research UK Cambridge Centre, University of Cambridge, Cambridge, United Kingdom

\* andrea.tangherloni@unibg.it (AT); daniela.besozzi@unimib.it (DB)

**Data Availability Statement:** The source code of FiCoS, both CPU and GPU versions, as well as all the scripts used to generate the results shown in the paper, are publicly and freely available on

## Abstract

Mathematical models of biochemical networks can largely facilitate the comprehension of the mechanisms at the basis of cellular processes, as well as the formulation of hypotheses that can be tested by means of targeted laboratory experiments. However, two issues might hamper the achievement of fruitful outcomes. On the one hand, detailed mechanistic models can involve hundreds or thousands of molecular species and their intermediate complexes, as well as hundreds or thousands of chemical reactions, a situation generally occurring in rule-based modeling. On the other hand, the computational analysis of a model typically requires the execution of a large number of simulations for its calibration, or to test the effect of perturbations. As a consequence, the computational capabilities of modern Central Processing Units can be easily overtaken, possibly making the modeling of biochemical networks a worthless or ineffective effort. To the aim of overcoming the limitations of the current state-of-the-art simulation approaches, we present in this paper FiCoS, a novel "black-box" deterministic simulator that effectively realizes both a fine-grained and a coarse-grained parallelization on Graphics Processing Units. In particular, FiCoS exploits two different integration methods, namely, the Dormand–Prince and the Radau IIA, to efficiently solve both non-stiff and stiff systems of coupled Ordinary Differential Equations. We tested the performance of FiCoS against different deterministic simulators, by considering models of increasing size and by running analyses with increasing computational demands. FiCoS was able to dramatically speedup the computations up to 855×, showing to be a promising solution for the simulation and analysis of large-scale models of complex biological processes.

GitLab: https://gitlab.com/andrea-tango/ficos. In the same repository, we also uploaded the code of the CPU versions of LSODA and VODE that were used to perform the tests shown in the paper. LASSIE and cupSODA source codes can be downloaded from the corresponding GitHub repositories. All other relevant data are within the manuscript and its Supporting information files.

**Funding:** This work was supported by the SYSBIO/ISBE.IT Centre of Systems Biology.

**Competing interests:** The authors have declared that no competing interests exist.

## Author summary

Systems Biology is an interdisciplinary research area focusing on the integration of biological data with mathematical and computational methods in order to unravel and predict the emergent behavior of complex biological systems. The ultimate goal is the understanding of the complex mechanisms at the basis of biological processes, together with the formulation of novel hypotheses that can be then tested by means of laboratory experiments. In such a context, mechanistic models can be used to describe and investigate biochemical reaction networks by taking into account all the details related to their stoichiometry and kinetics. Unfortunately, these models can be characterized by hundreds or thousands of molecular species and biochemical reactions, making their simulation unfeasible with classic simulators running on Central Processing Units (CPUs). In addition, a large number of simulations might be required to calibrate the models and/or to test the effect of perturbations. In order to overcome the limitations imposed by CPUs, Graphics Processing Units (GPUs) can be effectively used to accelerate the simulations of these models. We thus designed and developed a novel GPU-based tool, called FiCoS, to speed-up the computational analyses typically required in Systems Biology.

This is a *PLOS Computational Biology* Software paper.

## Introduction

The computational analysis of complex biological processes relies on the definition and simulation of mathematical models to investigate the emergent dynamics of these processes both in physiological and perturbed conditions [1]. The formalism based on Reaction-Based Models (RBMs) provides a detailed description of biochemical reaction networks [2], and represents one of the most suitable solutions to achieve an in-depth comprehension of the underlying control mechanisms of the cellular system under analysis [3, 4]. RBMs can be straightforwardly exploited to execute different computational tasks, such as Parameter Estimation (PE), Sensitivity Analysis (SA), and Parameter Sweep Analysis (PSA) [5–7], to gain insights about the system and drive the design of further laboratory experiments. However, these tasks typically require a huge amount of simulations of an RBM with distinct parameterizations, that is, different initial molecular amounts and/or kinetic constants. In addition, when the RBM under investigation is characterized by hundreds or thousands of molecular species and reactions, even one simulation could be so demanding that the capability of modern Central Processing Units (CPUs) is rapidly overtaken.

In the last decade, many efforts have been made to develop efficient computational tools able to perform massive numbers of simulations of large-scale RBMs. High-performance computing architectures, like Graphics Processing Units (GPUs), have gained ground and have been widely adopted for the parallelization of a variety of tasks in Systems Biology and Bioinformatics [8]. GPUs are low-cost, energy-efficient, and powerful many-core co-processors capable of tera-scale performances on common workstations, which can be exploited to dramatically reduce the running time required by traditional CPU-based approaches. Considering the GPU-powered biochemical simulation, the scheme in Fig 1 summarizes the current state-of-the-art. The existing tools can be categorized with respect to two main concepts: the simulation granularity, and the simulation type [8]. The former category determines how the threads

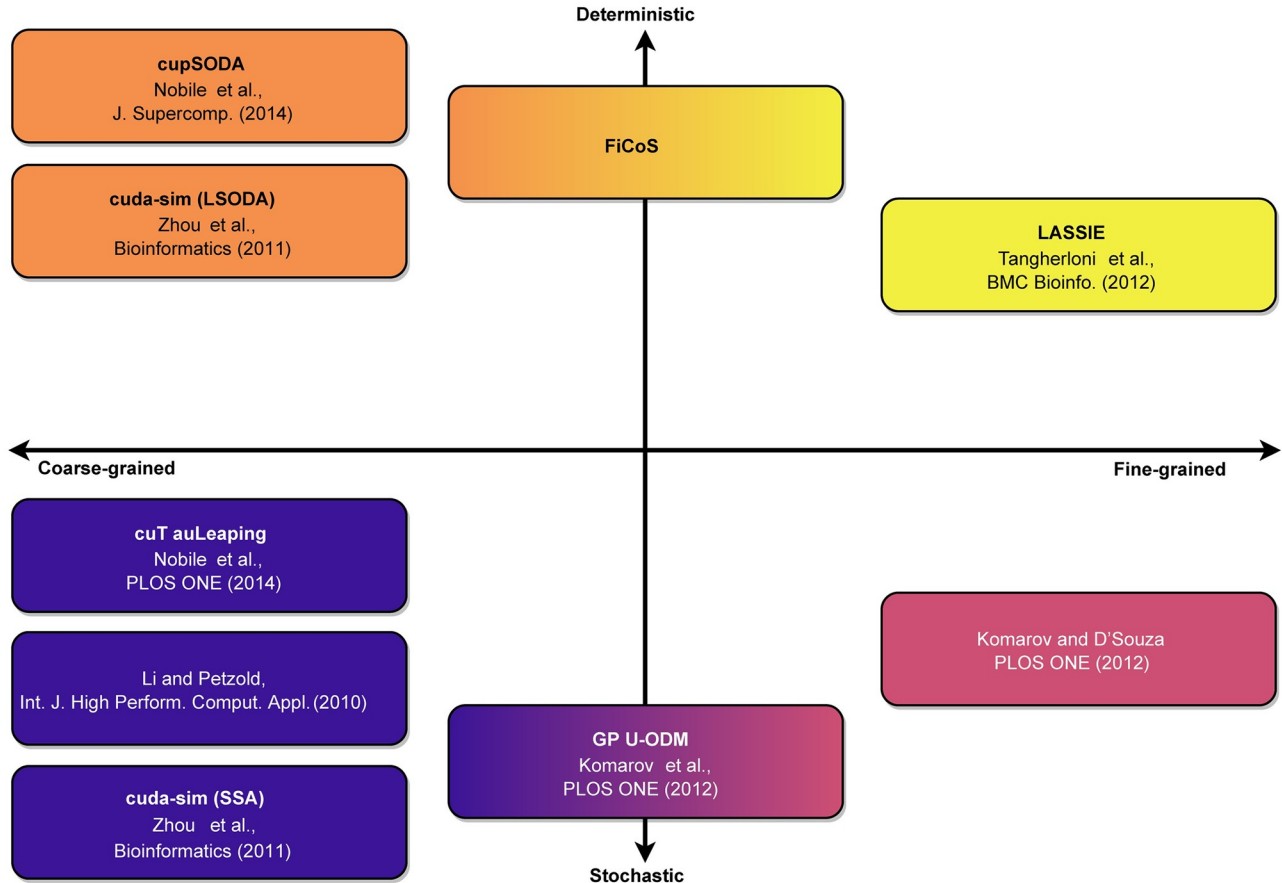

**Fig 1. The "semiotic square" of GPU-powered simulators.** The horizontal axis represents the parallelization strategy that can be realized to leverage the GPU capabilities: coarse-grained (left), in which multiple simulations are performed in a parallel fashion; fine-grained (right), in which a single simulation is accelerated by distributing the computation over multiple threads. The vertical axis partitions the two types of simulation: deterministic (top) and stochastic (bottom). FiCoS is a deterministic approach implementing both a fine- and a coarse-grained parallelization strategy, filling the gap in the state-of-the-art.

of the GPU can be used: in coarse-grained simulations, each thread corresponds to an independent simulation; in fine-grained simulations, the calculations of a single simulation are distributed over multiple threads. The latter category specifies whether a deterministic, stochastic, or hybrid simulation algorithm is used [9–11].

The first GPU-powered simulator was proposed by Li and Petzold [12], and consisted in the implementation of the Stochastic Simulation Algorithm (SSA) [13]. SSA was later implemented in cuda-sim [14], a Python tool for Systems Biology that also supports deterministic simulations. Nobile *et al.* proposed two further GPU-based coarse-grained implementations: cupSODA [15], a deterministic simulator based on the Livermore Solver of Ordinary Differential Equations (LSODA) [16], and cuTauLeaping [17], a stochastic simulator based on a variant of the tau-leaping algorithm [18]. cupSODA was then integrated with PySB [19], a well-known modeling and simulation framework that allows the user to create biological models as Python programs [20]. To date, there exist only two purely fine-grained approaches: LASSIE (LArge-Scale SImulator) [21], a fine-grained deterministic simulator tailored on large-scale models (i.e., with more than 100 chemical species), and the large-scale tau-leaping variant proposed by Komarov and D'Souza [22], designed to accelerate very large-scale stochastic models (i.e., with more than $10^5$ reactions). Two mixed approaches of fine- and coarse-grained acceleration are

GPU-ODM [23], which is based on a variant of SSA called the Optimized Direct Method, and the SSA proposed by Sumiyoshi *et al.* [24]. Besides these GPU-powered tools, other solutions have been recently proposed, such as the Stochastic Simulation as a Service (StochSS) [25], and the Systems Biology Simulation Core Library (SBSCL) [26]. StochSS is an integrated development environment that can be used to model and simulate both deterministic and discrete stochastic biochemical systems in up to three dimensions. SBSCL provides interpreters and solvers for different Systems Biology formats, like the Systems Biology Markup Language (SBML) [27]. SBSCL allows the user to simulate several models by supporting the resolution of Ordinary Differential Equations (ODEs) and Stochastic Differential Equations.

In this work, we introduce FiCoS (**Fi**ne- and **Co**arse-grained **S**imulator), an efficient deterministic simulator that, differently from the state-of-the-art approaches, was designed to fully exploit the parallelism provided by modern GPUs, by using a combined fine- and coarse-grained parallelization strategy. In particular, the fine-grained parallelization is applied to distribute the calculations required to solve each ODE over multiple GPU cores, while the coarse-grained parallelization is used to perform batches of independent simulations in parallel. To the best of our knowledge, FiCoS is the first deterministic simulator that takes advantage of both parallelization strategies, filling the gap in the state-of-the-art scenario of GPU-powered deterministic simulators (Fig 1).

FiCoS improves the approach proposed in LASSIE [21] by leveraging more efficient integration methods, i.e., the Dormand–Prince (DOPRI) method [28–30] for stiffness-free systems, and the Radau IIA method [31, 32] when the system of ODEs is stiff. A system is said to be stiff when two well-separated dynamical modes, determined by fast and slow reactions, occur: after a short transient the fast modes lead to stable trajectories, where only the slow modes drive the dynamics [33, 34]. As a matter of fact, systems of ODEs describing biochemical networks are often affected by stiffness [35], an issue that generally requires the reduction of the step-size of explicit integration algorithms up to extremely small values, thus increasing the running time.

The performance of FiCoS was assessed by means of a batch of tests involving up to 2048 parallel simulations of (*i*) synthetic RBMs of increasing size (up to hundreds of species and reactions); (*ii*) a model describing the autophagy/translation switch based on the mutual inhibition of MTORC1 and ULK1 [36], characterized by 173 molecular species and 6581 reactions; (*iii*) a model of human intracellular metabolic pathways consisting in 226 reactions involving 114 molecular species [37].

The running time required by FiCoS to perform these analyses was compared to the CPU-based ODE solvers LSODA and Variable-Coefficient ODE (VODE) [38], which are exploited by several software for Systems Biology. For instance, libRoadRunner [39] uses CVODE [40] (a porting of VODE in the C programming language) and a standard fourth-order Runge–Kutta method; COPASI [41] performs deterministic simulations using LSODA and an implementation of the RADAU5 method (included only in the latest versions); VCell [42] exploits CVODE and other simpler integration algorithms. In this work, we measured the performance of FiCoS against vanilla ODE solvers, rather than considering a specific software, to achieve a fair comparison. By so doing, we can separately measure the time required by the integration methods and the time spent by the simulators for I/O operations. In particular, the CPU-based ODE solvers employed here are those implemented in the Python SciPy library, which provides wrappers to the solvers implemented in the ODEPACK, a collection of efficient Fortran solvers. Moreover, the performance of FiCoS was compared against those of the GPU-powered simulators cupSODA [15] and LASSIE [21], implemented using the Nvidia CUDA library [43].

Our results show that FiCoS represents the best solution in the case of computational tasks involving either a high number of simulations or large-scale models, overcoming the performance of all state-of-the-art competitors.

## Results

In this section, the computational performance of FiCoS is compared against state-of-the-art ODE solvers, that is, LSODA [16], VODE [38], cupSODA [15], and LASSIE [21]. LSODA [16] is based on two families of multi-step methods, namely, the Adams methods [44], which are explicit integration algorithms inefficient for solving stiff systems, and the Backward Differentiation Formulae (BDF) [45], which are implicit integration methods suitable to solve stiff systems. LSODA can efficiently solve stiff systems by switching among the most appropriate integration algorithm during the simulation. Conversely, despite being based on the same integration algorithms used by LSODA, VODE exploits, at the beginning of the simulation, a heuristic to decide the best algorithm that should be used to integrate the system of ODEs. LASSIE solves the systems of ODEs by automatically switching between the Runge-Kutta-Fehlberg method in the absence of stiffness, and the first order BDF in presence of stiffness. FiCoS exploits a heuristic to determine if the system is stiff and which integration method, between DOPRI and Radau IIA, is the most appropriate (see S1 Text).

We performed two batches of tests using synthetic RBMs of increasing size (i.e., number of species and number of reactions), generated as described in the Section "Materials and methods". To provide some examples of real Systems Biology tasks, we also carried out an in-depth investigation of the RBMs describing the autophagy/translation switch based on mutual inhibition of MTORC1 and ULK1 [36], and a human intracellular metabolic pathway [37]. All tests were executed on a workstation equipped with an Intel Core i7–2600 CPU (clock 3.4 GHz) and 8 GB of RAM, running Ubuntu 16.04 LTS. The GPU used in the tests was an Nvidia GeForce GTX Titan X (3072 cores, clock 1.075 GHz, RAM 12 GB), CUDA toolkit version 8 (driver 387.26). All simulations were performed with the following settings: absolute error tolerance $\varepsilon_a = 10^{-12}$, relative error tolerance $\varepsilon_r = 10^{-6}$, and maximum number of allowed steps equal to $10^4$ (these values are widely used in the literature, as well as by the main computational tools like COPASI [41]). In all tests, we leveraged the LSODA and VODE implementations provided by the SciPy scientific library (v.1.4.1) [46] along with the NumPy library (v.1.18.2) under Python (v.3.8.5). In all the simulations, we saved the dynamics of all species of the RBMs.

### Synthetic models

The computational analyses carried out on synthetic models aimed at determining what is the best simulator to employ under specific conditions, i.e., considering computational tasks with an increasing number of parallel simulations (up to 2048) and different RBM sizes (from tens to hundreds or even thousands of chemical species and reactions).

The map in Fig 2 shows the results obtained in the case of symmetric RBMs, i.e., having the same number of chemical species and reactions ($N = M$). When a single simulation is performed, the CPU-based solvers (i.e., LSODA and VODE) achieved the best performance in the case of RBMs smaller than $N \times M = 512 \times 512$, which is the break-even point between CPU- and GPU-based simulators. Indeed, when larger models are considered, FiCoS and LASSIE are the best choices, thanks to their parallelization strategies that offload the calculations onto the GPU. Considering the case of multiple independent simulations, our results indicate that cupSODA should be adopted with small-scale RBMs (i.e., less than 128 species and reactions) when the execution of a limited number of parallel simulations is required (i.e., less than 256). Under these circumstances, cupSODA can leverage the most performing GPU memories, that

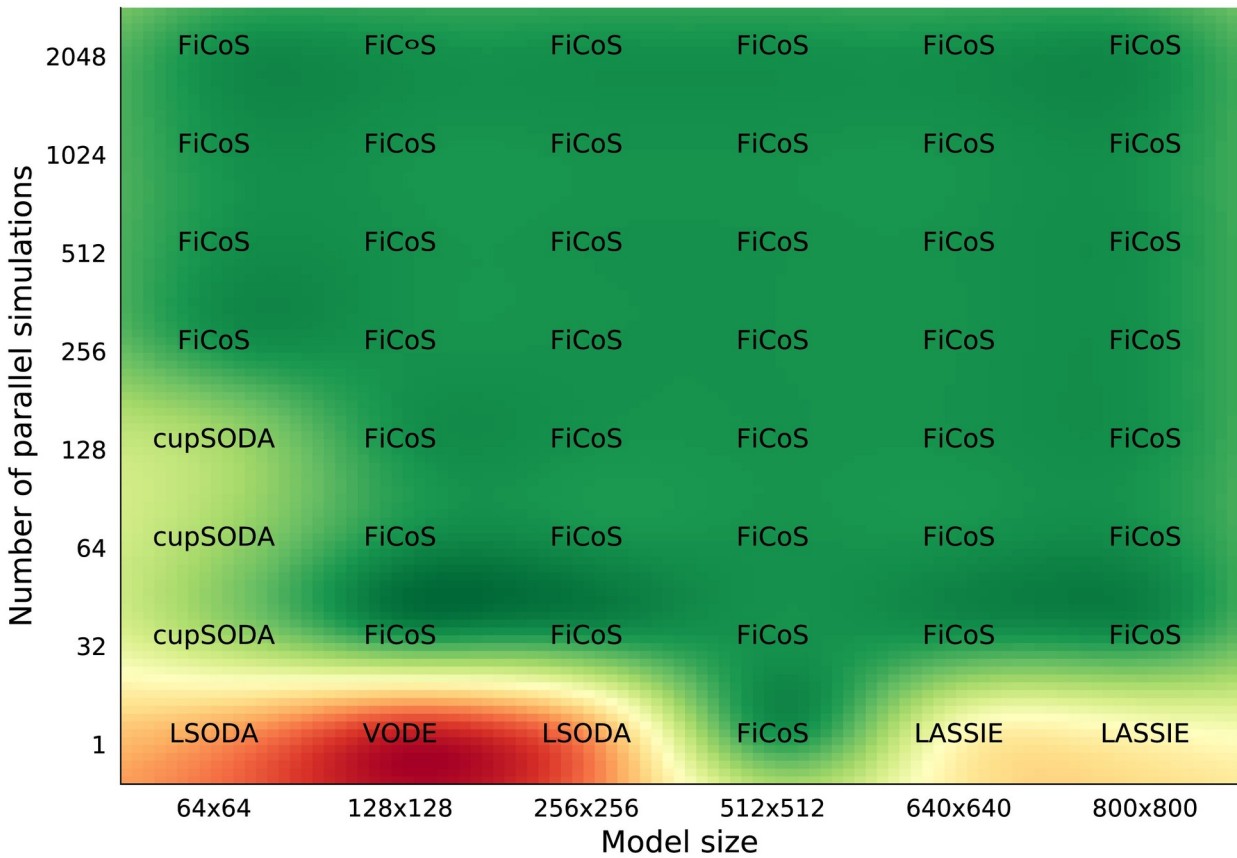

**Fig 2. Comparison map for the symmetric RBMs.** This map represents the best simulator in terms of running time required to simulate symmetric RBMs of size $N \times M$, with $N = M$, where $N$ is the number of species and $M$ the number of reactions. The running time was analyzed increasing both the size of the RBMs and the number of simulations.

is, the constant and the shared memories, which provide a relevant reduction of memory access latencies. As shown by the green area in Fig 2, FiCoS outperforms all its competitors in all the other cases thanks to its fine- and coarse-grained parallelization strategy, which allows for simultaneously distributing both the parallel simulations required by the specific computational task and the ODE calculations on the available GPU cores.

It is worth noting that the performance of the simulators can drastically change when dealing with RBMs characterized by a different number of chemical species and reactions (i.e., asymmetric RBMs, where $N \neq M$). As a matter of fact, since each ODE corresponds to a specific chemical species, the higher the number of species, the higher the parallelization that can be achieved by exploiting the fine-grained strategy. On the contrary, the number of reactions is roughly related to the length (in terms of mathematical complexity) of each ODE, thus increasing the number of operations that must be performed by each thread. In addition, since the clock frequency of the GPU cores is generally lower than that of the CPU, the time required to perform a single instruction on the GPU is higher. For this reason, the performance that can be achieved on the GPU decreases when the number of reactions $M$ is larger than the number of species $N$. Figs 3 and 4 show the results of the tests performed on asymmetric RBMs with $N > M$ and $M > N$, respectively. We first observe that the previous results are confirmed, as GPU-powered simulators most of the times overcome the CPU-based simulators. In particular, in the case of a single simulation of RBMs with $N > M$ (Fig 3), LSODA is efficient

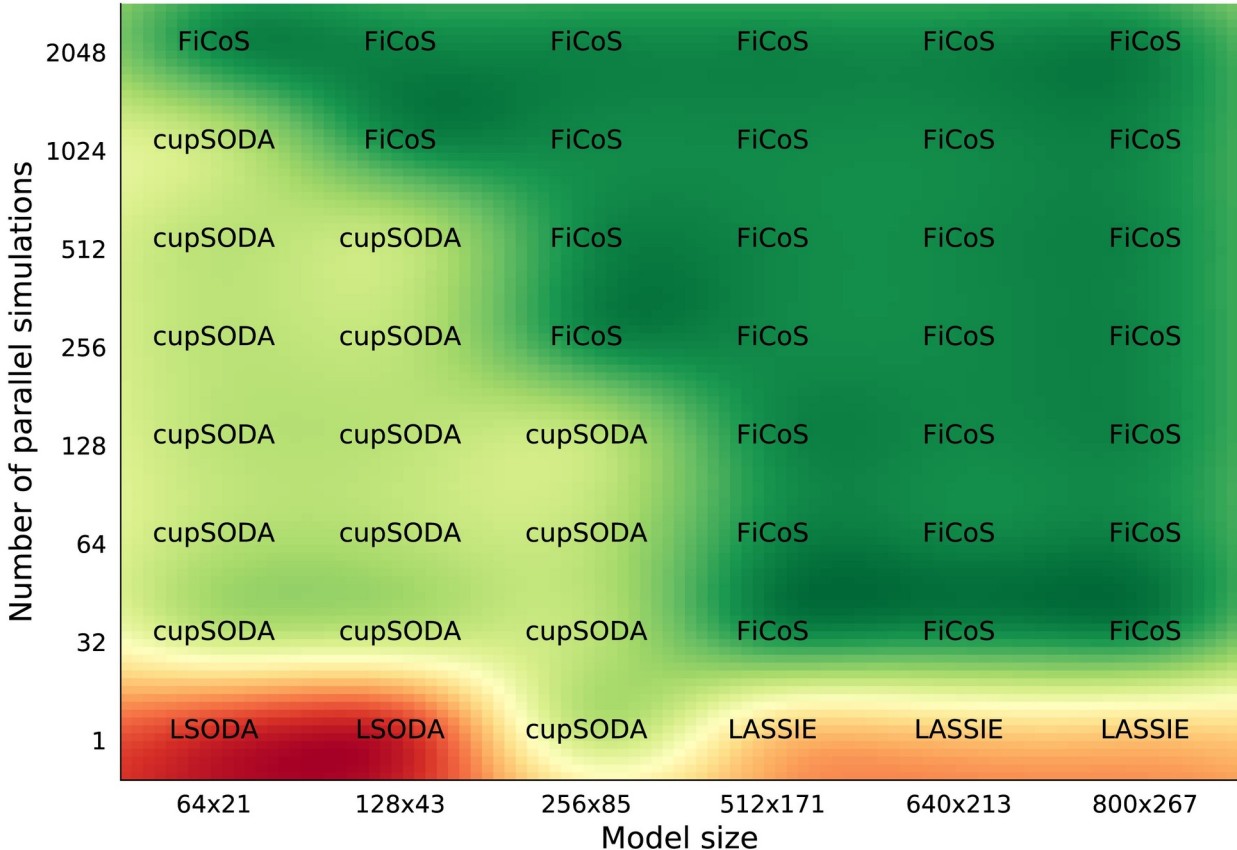

**Fig 3. Comparison map for the asymmetric RBMs with more species than reactions.** This map represents the best simulator in terms of running time required to simulate asymmetric RBMs of size $N \times M$, with $N > M$, where $N$ is the number of species and $M$ the number of reactions. The running time was analyzed increasing both the size of the RBMs and the number of simulations.

only for RBMs with a limited number of chemical species ($N \leq 128$), while cupSODA and LASSIE represent the best option for large-scale RBMs. If the number of simulations is increased, either cupSODA or FiCoS should be employed: the former for small-scale RBMs, the latter in all other cases. Considering the case of RBMs with $M > N$ (Fig 4), the best options for the single simulation are LSODA for RBMs with size up to $213 \times 640$, and FiCoS otherwise. Also in this case, if the number of simulations is increased, cupSODA and FiCoS are the best solutions for small-scale and large-scale RBMs, respectively. These results confirm that the number of reactions affects the performance that can be achieved by using GPUs; as a matter of fact, cupSODA is the best solution only for RMBs with size $21 \times 64$, while FiCoS should be adopted in all other cases.

Overall, our analyses prove the relevance of GPU-powered simulators, especially when high numbers of simulations must be performed or large-scale RBMs are used.

## Autophagy/translation model

In order to show the advantages provided by FiCoS in the investigation of models of real biological systems, we performed a bi-dimensional PSA (PSA-2D) on the mathematical model of the autophagy/translation switch based on the mutual inhibition of MTORC1 and ULK1 presented in [36]. These two proteins are part of a large regulatory network that is responsible of maintaining cellular energy and nutrients' homeostasis. Autophagic processes are involved in

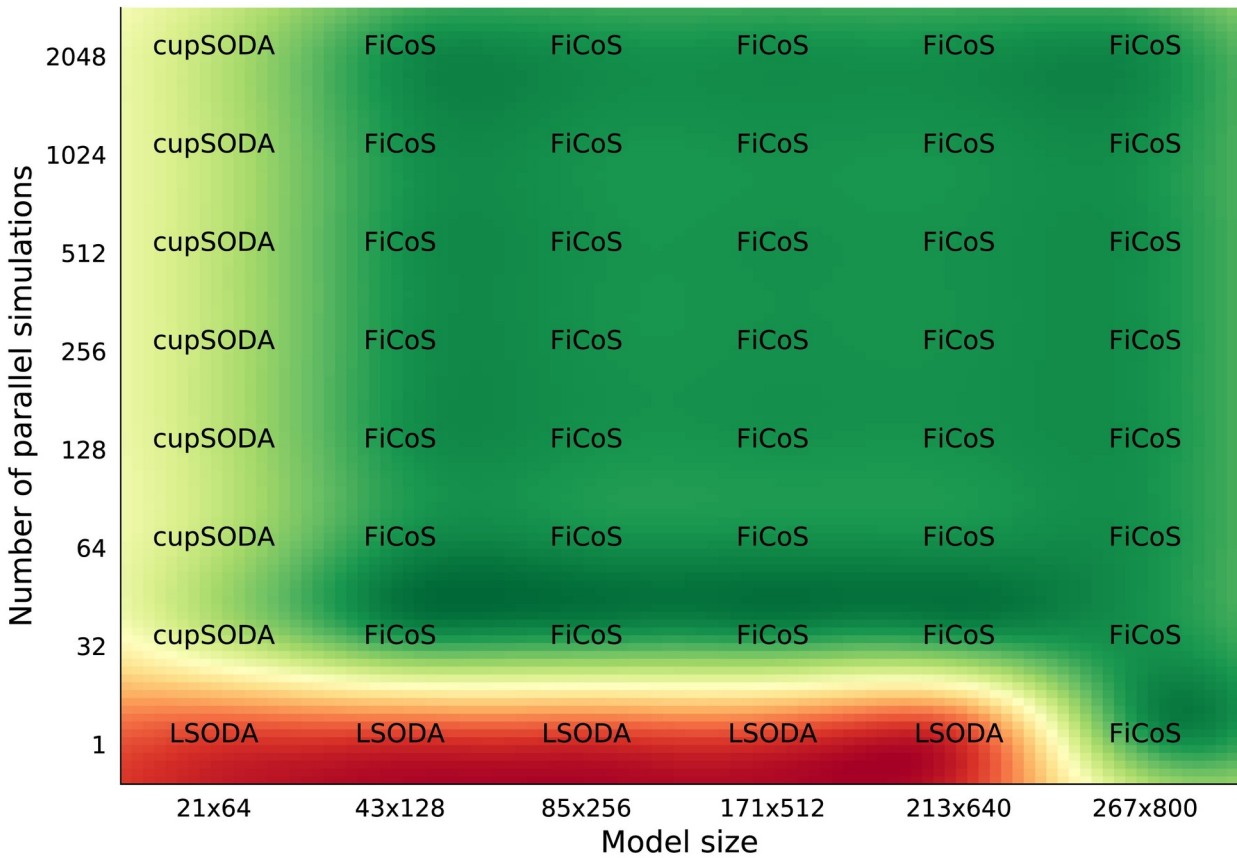

**Fig 4. Comparison map for the asymmetric RBMs with more reactions than species.** This map represents best simulator in terms of running time required to simulate asymmetric RBMs of size $N \times M$, with $N < M$, where $N$ is the number of species and $M$ the number of reactions composing the models. The running time was analyzed increasing both the size of the RBMs and the number of simulations.

cell survival during starvation and clearing of damaged cellular components, and thus their investigation is fundamental to understand aging-related and neurodegenerative diseases, as well as immunity and tumorigenesis.

The PSA-2D presented hereafter was already performed in [36] to study the oscillations in the phosphorylation levels of AMBRA1, a protein responsible for the activation of autophagy, and EIF4EBP1, a repressor of translation. Such oscillations generate alternating periods of autophagy and translation inside the cell, in response to varying levels of phosphorylated AMPK (AMPK*), whose presence represents a stress condition for the cell, and of the parameter $P_9$, which models the strength of the negative regulation of MTORC1 by AMPK*.

The mathematical model of the autophagy/translation switch was defined using BioNetGen rule-based modeling language (BNGL) [47]. Rule-based modeling is a practical means to describe, in a concise way, the interactions between molecules while keeping track of site-specific details (e.g., the phosphorylation states). We simulated the rule-based model by deriving the corresponding network of reactions, written as a conventional RBM. In particular, the rule-based model is characterized by 7 initial molecule types, involved in 29 rules; the associated RBM contains 173 molecular species and 6581 reactions.

The PSA-2D was performed by varying simultaneously the two parameters that affect the oscillatory behavior, that is, the initial amount of AMPK* and the value of the parameter $P_9$, which modifies 5476 kinetic constants of the corresponding RBM. The initial values of both

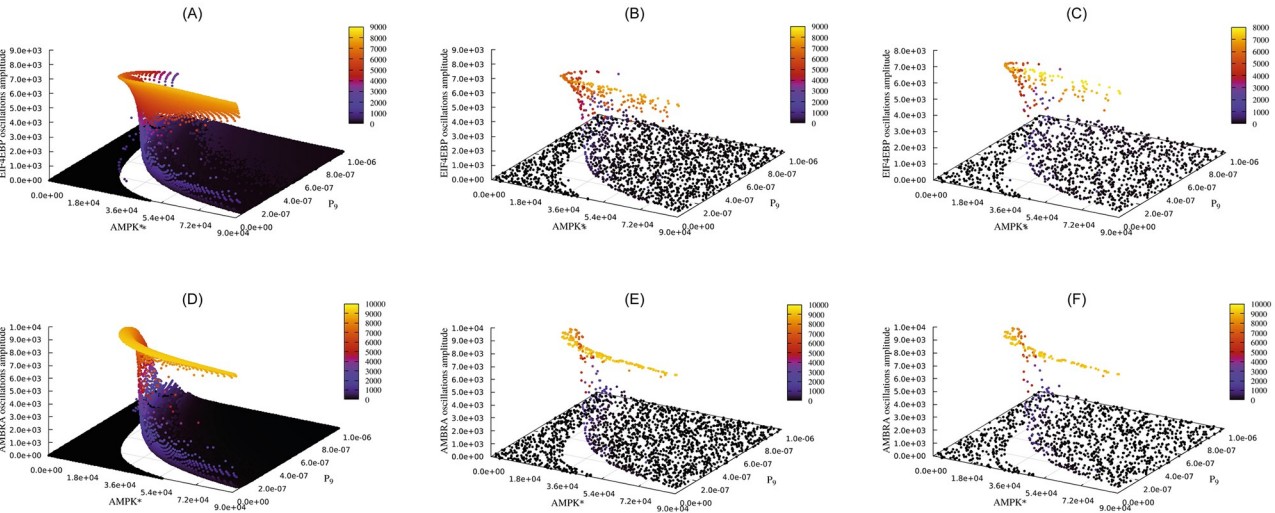

**Fig 5. Results of the PSA-2D performed on the autophagy/translation model.** This PSA-2D was realized by varying the initial concentration of AMPK* and the value of the parameter $P_9$, which modifies 5476 kinetic parameters. The plots show the average oscillations amplitude of the species EIF4EBP and AMBRA. (A) Oscillations amplitude of EIF4EBP obtained by FiCoS. (B) Oscillations amplitude of EIF4EBP obtained by LSODA. (C) Oscillations amplitude of EIF4EBP obtained by VODE. (D) Oscillations amplitude of AMBRA obtained by FiCoS. (E) Oscillations amplitude of AMBRA obtained by LSODA. (F) Oscillations amplitude of AMBRA obtained by VODE. Considering a time budget of 24 hours, FiCoS was capable of performing 36864 simulations, while LSODA and VODE performed 2090 and 1363 simulations, respectively.

parameters were obtained by uniformly sampling within the sweep intervals proposed in the original paper [36], namely: $[0, 10^4]$ (molecules/cell) for the initial amount of AMPK*, and $[10^{-9}, 10^{-6}]$ ((molecules/cell)$^{-1}$s$^{-1}$) for the value of $P_9$. We performed a total of 36864 simulations with FiCoS, by running 72 batches of 512 simulations, a value that allowed us to maximize the performance of the simulator on the GPU, as evidenced by specific computational tests. FiCoS completed all simulations in approximately 24 hours; in the same time span, LSODA and VODE were capable of performing only 2090 and 1363 simulations, respectively.

Fig 5 shows the average amplitude of EIF4EBP and AMBRA oscillations obtained from the PSA-2D using FiCoS (Fig 5A and 5D), LSODA (Fig 5B and 5E), and VODE (Fig 5C and 5F). Amplitude values equal to zero (black points) denote a non-oscillating dynamics of the molecular species. As it can be observed in the figure, although some information about the behavior of the system can be extrapolated by the results achieved with LSODA and VODE, FiCoS is the only simulation strategy that can provide biologists with extremely detailed analyses of complex cellular models in a limited time span.

It is worth noting that we compared FiCoS only against LSODA and VODE for the following motivations: (*i*) LSODA and VODE are the most used simulators; (*ii*) the tests on the asymmetric synthetic RBMs with more reactions than species showed that the performance of LASSIE is much lower than FiCoS (see S1 Text); (*iii*) cupSODA is not suitable for the simulation of large-scale RBMs, as its memory requirements prevent the execution of this kind of tasks.

## Human intracellular metabolic pathway

As a further example of Systems Biology tasks that can be efficiently tackled with FiCoS, we performed an SA and a PE of an RBM of the human intracellular metabolic pathway in red blood cells, presented in [48]. The basic version of this model is composed of 94 reactions involving 92 chemical species, which describe the central pathways of the carbohydrate

metabolism (i.e., glycolysis and pentose phosphate pathways). Here, we consider the modified model version presented in [37], in which the uptake of extracellular substrates is not considered, while the isoforms of the hexokinase (HK) enzyme, fundamental to convert the glucose into glucose-6-phosphate, are explicitly taken into account. The resulting RBM is characterized by 226 reactions among 114 chemical species.

Metabolic networks can be considered as complex systems, characterized by hundreds of reactions and chemical species, where intra- and extra-cellular processes are involved [49, 50]. In these networks, several sources of indetermination can hamper the outcome of the analyses, especially when different uncharacterized enzyme isoform mixtures exist [51]. Enzyme isoforms (i.e., isozymes) can be seen as non-identical, albeit structurally similar, protein complexes that are able to catalyze the same biochemical reactions. However, the structural differences of these isoforms lead to different kinetic behaviors of the catalytic processes [52, 53].

The SA was performed in order to evaluate the effects of an isoform-specific modification on the system outcome, by varying the initial concentration of the HK isoform with the highest abundance. In a real scenario, the modification of an isoform can be obtained with a tailored gene knock-down experiment, by means of a drug with an isoform-specific target, or with a change in the isozyme expression by exposing the cells to an environmental stimulus [50]. Here, we modified both the initial concentration of this HK isoform, as well as that of all the intermediate chemical complexes produced by this HK isoform. We sampled the initial concentration of these 11 species in the range $[0, 10^{-5}]$ (mM).

We exploited the Sobol method for SA [54, 55], which is a variance-based SA where the variance of the model output is decomposed into fractions that can be attributed to the inputs, across the whole input space, and it is also capable of dealing with nonlinear responses. We applied the Saltelli sampling scheme [56], an extension of the Sobol sequence that aims at reducing the error rates in the resulting sensitivity index calculations. To calculate the first- and the second-order sensitivities, we considered 12288 distinct initial conditions obtained by sampling 512 points for each of the 11 species involved in the SA. We simulated the dynamics of the ribose 5-phosphate (R5P) metabolite, which is involved in the pentose phosphate pathway, in a 10-hour time window; then, we calculated the difference between the final concentration value achieved and the reference value (obtained by running a simulation with the initial conditions listed in [37]). These values were finally used to compute the SA outcome, reported in Table 1, which lists the first- and total-order sensitivity indices, together with the

**Table 1. First- and total-order indices along with the corresponding confidence intervals (confidence level of 95%) of the Sobol SA, calculated on the R5P metabolite outputs.**

| Species | $S_1$ | $S_{1_{conf}}$ | ST | $ST_{conf}$ |
|---|---|---|---|---|
| $hkE_2$ | −0.005 | 0.046 | 0.138 | 0.020 |
| $hkEMgATP_2$ | 0.016 | 0.052 | 0.139 | 0.019 |
| $hkEMgATPGLC_2$ | 0.008 | 0.053 | 0.139 | 0.019 |
| $hkEGLC_2$ | 0.041 | 0.051 | 0.148 | 0.025 |
| $hkEMgADPG6P_2$ | 0.031 | 0.046 | 0.145 | 0.023 |
| $hkEG6P_2$ | −0.010 | 0.050 | 0.153 | 0.026 |
| $hkEMgADP_2$ | 0.029 | 0.044 | 0.112 | 0.022 |
| $hkEGLCGSH_2$ | 0.199 | 0.078 | 0.487 | 0.084 |
| $hkEGLCDPG23_2$ | 0.240 | 0.079 | 0.462 | 0.075 |
| $hkEPhosi_2$ | 0.233 | 0.085 | 0.509 | 0.086 |
| $hkEGLCG6P_2$ | 0.212 | 0.081 | 0.482 | 0.090 |

corresponding confidence intervals, considering a confidence level of 95%. These results clearly show that the modification of the initial concentrations of the species $hkEGLCGSH_2$, $hkEGLCDPG23_2$, $hkEPhosi_2$, and $hkEGLCG6P_2$ have the largest impact on the dynamics of the R5P metabolite.

Concerning the running times, FiCoS was able to perform the 12288 simulations—divided into 24 batches of 512 simulations (i.e., the value that maximizes the performance of FiCoS)—in approximately 8 minutes, while LSODA only completed 103 simulations in the same time span. Thus, FiCoS was $\sim 119\times$ faster than LSODA.

As shown in the extended version of this model of the human intracellular metabolic pathway [37], the values of 78 out of 226 kinetic constants are unknown and need to be estimated to perform a correct simulation of the dynamics. We thus performed a PE of these 78 unknown values, following the approach described in [37]. In particular, we compared the running times of the PE executed with the Fuzzy Self-Tuning PSO (FST-PSO) algorithm [57], a setting-free version of Particle Swarm Optimization [58] that was first coupled with FiCoS and then with LSODA. It is worth noting that in this PE approach, each particle of the FST-PSO encodes a putative model parameterization, whose quality is evaluated by means of a fitness function that computes the relative distance between the target dynamics and the simulated dynamics obtained using that putative parameterization [7, 59]. FiCoS allowed for efficiently running the simulations required in the PE to assess the quality of the solutions encoded in the particles of FST-PSO; indeed, FiCoS was able to execute the PE $\sim 30\times$ faster than LSODA, showing again the high relevance of GPU-powered simulators for the execution of these computationally demanding tasks in practical applications.

## Discussion

The emerging field of General Purpose Computing on Graphics Processing Units (GPGPU computing)—in which CPUs and GPUs are combined to address the increasing demand of parallel and throughput-intensive algorithms—had a relevant impact to many fields in both science and engineering [60]. Specifically, GPGPU computing enabled the development of novel software tools that have been successfully applied to tackle the huge computational efforts required by several disciplines related to life sciences [8]. The accelerated biochemical simulation method proposed here, named FiCoS, represents the first open-source deterministic simulator that takes advantage of both fine- and coarse-grained strategies to distribute the massive amount of calculations required by multiple simulations of large-scale RBMs. FiCoS can be effectively applied to tackle the demanding tasks that are generally required in Computational Systems Biology, as in the case of the PSA, SA, and PE presented in this work. The source-code of FiCoS is freely available on GitLab: https://gitlab.com/andrea-tango/ficos under the BSD 3-Clause License. This repository also contains a description of all input parameters of FiCoS, as well as all the scripts used to perform the computational analyses presented in this work and Jupyter Notebooks with detailed information to run PSA-2D, SA, and PE with FiCoS.

In terms of performance, FiCoS is capable of solving systems of non-stiff and stiff ODEs with similar and often higher precision and accuracy than other well-known ODE solvers (see Section "Simulation accuracy of FiCoS" in the S1 Text), but with a dramatic reduction of the execution time. In our tests, FiCoS resulted faster than CPU-based integration methods like LSODA and VODE when multiple simulations need to be executed. Moreover, in many cases FiCoS shown to be faster than LASSIE, which was designed to exploit the fine-grained strategy to distribute the calculations required by a single simulation of large-scale RBMs. As shown in S1 Text (Section "Computational performance"), FiCoS achieves relevant speed-ups,

resulting up to 487× faster than VODE in terms of simulation time and up to 855× considering only the integration time; up to 366× faster than LSODA for the simulation time and up to 79× faster for the integration time only. Considering the comparison with the other GPU-based simulators, FiCoS is 298× faster than LASSIE in terms of simulation time and 760× faster for the integration time only, thanks to the double level of parallelism and the more efficient integration methods employed. FiCoS was also capable of outperforming cupSODA, the GPU implementation of LSODA that exploits a coarse-grained parallelization strategy, resulting 7× faster in terms of simulation time and 17× considering only the integration time. To clarify, the integration time indicates the running time spent by a numerical integration algorithm to solve the system of ODEs, while the simulation time is the overall running time required to perform a simulation, including the I/O operations (i.e., reading and writing operations).

The analyses presented in this paper allowed us to also identify the limitations of FiCoS. First, since FiCoS was specifically designed to simulate large-scale RBMs, the intra-GPU communication overhead—caused by the fine-grain kernels—affects its performance in the case of small size models. Moreover, as expected, when only a few parallel simulations are required, other CPU- or GPU-based simulators might be preferred. Second, the performance of FiCoS related to the coarse-grained strategy can be affected by the saturation of the computing resources due to an "excessive" parallelization. This phenomenon is caused by the computing resources required by the kernels realizing the fine-grained strategy—exploited to parallelize the resolution of the system of ODEs—which increase along with the RBM size. As a matter of fact, the blocks of threads generated using the Dynamic Parallelism (DP) [43, 61] (see S1 Text for details) easily saturate the GPU resources and reduce the speed-up. As shown in [62], the launching time of child kernels slightly increases after 512 kernels, and dramatically increases around 2000 kernels. Note that the parent grid is launched by the host, while a child grid (composed of child blocks, threads, and kernels) can be launched from each parent thread, resulting in the parallel execution of several child kernels. In addition, the register usage raises along with the number of threads running in parallel, decreasing the performance if the execution requires the access to the global memory, which has high latencies. According to our tests, the performance of FiCoS decreases when more than 2048 parallel simulations of large-scale RBMs are executed.

## Conclusion

In this work we proposed FiCoS, a GPU-powered deterministic simulator, designed to deal with the computational demanding tasks generally required in the field of Computational and Systems Biology. FiCoS is capable of simulating large-scale RBMs based on mass-action kinetics relying on fine- and coarse-grained parallelization strategies.

FiCoS was designed to be a "black-box" simulator able to automatically convert RBMs, representing complex biochemical networks, into the corresponding systems of ODEs. These systems are solved by exploiting two Runge-Kutta methods, that is, the DOPRI method [28–30] in the absence of stiffness and the Radau IIA method [31, 32] when the system is stiff. This mixed parallelization strategy that takes advantage of the DP provided by modern GPUs allows for exploiting the massive parallel capabilities of GPUs, which are essential to obtain a relevant reduction of the computation time.

The performance of FiCoS was evaluated considering two sets of synthetic RBMs of increasing size (up to 800 molecular species and reactions), a real model of the autophagy/translation switch, based on the mutual inhibition of MTORC1 and ULK1, characterized by 173 molecular species and 6581 reactions, and a human intracellular metabolic pathway, composed of 226

reactions and 114 molecular species. We performed an in-depth analysis to determine the best simulator that should be employed considering both the size of the RBM and the number of simulations required. In particular, we compared the performance of FiCoS against two CPU-based simulators (LSODA and VODE), a fine-grained GPU-powered simulator designed to simulate large-scale models (LASSIE), and a coarse-grained GPU-powered simulator developed to run high numbers of simulations in a parallel fashion (cupSODA). Exploiting both the fine- and the coarse-grained parallelization strategies, FiCoS resulted the natural choice when multiple simulations are required, and when the number of species in the RBM is greater than 64, noticeably outperforming all the other simulators. We also performed a bi-dimensional PSA of a model of the autophagy/translation switch based on the mutual inhibition of MTORC1 and ULK1 [36], showing that FiCoS was able to execute 36864 simulations in 24 hours, while LSODA and VODE completed only 2090 and 1363 simulations in the same time span, respectively. Moreover, we performed an SA of a human intracellular metabolic pathway with different enzyme isoforms, requiring the execution of 12288 simulations. FiCoS completed this task in approximately 8 minutes, while LSODA was able to perform only 103 simulations in the same time span. Finally, we coupled FST-PSO [57] with both FiCoS and LSODA to run a PE of the RBM of a human intracellular metabolic pathway. Our analysis showed that FiCoS allows for performing this task $\sim 30\times$ faster than LSODA.

Although FiCoS showed excellent performance, we did not fully exploit the CUDA memory hierarchy that should be used as much as possible to obtain the best performance. Since both explicit and implicit integration algorithms are intrinsically sequential, FiCoS kernels generally do not reuse any variable, so that the (fast) shared memory of the GPU cannot be leveraged. Moreover, the DP does not currently allow for sharing variables among the threads belonging to the parent grid and those inside the child grids. For these reasons, the current version of FiCoS relies on the global memory (characterized by high latencies) and registers. We plan to develop an improved version of FiCoS tailored for RBMs characterized by a few number of species, leveraging both constant and shared memories, where the former can be used to store the kinetic constants and the structures used to correctly decode the ODEs, while the latter can be exploited to save the states of the system.

Another limitation of FiCoS regards its capability of dealing with RBMs including reactions following kinetics different than mass-action. This limitation arises from the way ODEs are encoded by FiCoS and are parsed by the CUDA kernels. We are therefore planning to develop an improved version of this encoding able to represent Hill and Michaelis-Menten kinetics [63, 64], in addition to mass-action kinetics. However, we expect that the new parser will slightly affect the performance of FiCoS, mainly because of an increased level of conditional branching and higher latencies due to additional global memory accesses. We are also evaluating the possibility of defining a general-purpose version of FiCoS, capable of simulating models where the reactions follow any arbitrary kinetics. In this case, the main issue would concern the complexity of encoding arbitrary equations and the resulting difficulty of calculating partial derivatives for the Jacobian matrix. A possible solution would be meta-programming, following a strategy similar to ginSODA [65], in which the kernels are partially rebuilt and dynamically linked at run-time. However, such an approach is unfeasible for fine-grained parallelization, thus it should be adapted to be applied in FiCoS.

As a final remark, since several existing tools, such as libRoadRunner, COPASI, and VCell exploit the SBML format [27] to encode RBMs, while FiCoS, LASSIE, and cupSODA rely on the BioSimWare format [66], we provided in the GitLab repository a simple conversion tool to translate an SBML model into the corresponding BioSimWare model. We are currently working on an improved version of this conversion tool, capable of translating BioSimWare files to SBML files, taking into account all model parameterizations; as a matter of fact, the

BioSimWare format allows for easily specifying multiple parameter sets and initial conditions that might be tested in practical applications.

## Materials and methods

### RBMs and ODE generation

RBMs are defined following a mechanistic, quantitative, and parametric formalism created to describe and simulate networks of biochemical reactions [2]. An RBM can be defined by specifying the set $\mathcal{S} = \{S_1, \ldots, S_N\}$ of $N$ molecular species and the set $\mathcal{R} = \{R_1, \ldots, R_M\}$ of $M$ biochemical reactions that describe the interactions among the species in $\mathcal{S}$. A generic reaction $R_i$, with $i = 1, \ldots, M$, is defined as follows:

$$R_i : \sum_{j=1}^{N} a_{ij} S_j \xrightarrow{k_i} \sum_{j=1}^{N} b_{ij} S_j, \quad i = 1, \ldots, M, \tag{1}$$

where $a_{ij}, b_{ij} \in \mathbb{N}$ are the so-called stoichiometric coefficients and $k_i \in \mathbb{R}^+$ is the kinetic constant associated with $R_i$. Note that $\mathcal{R}$ can be also written in a compact matrix-vector form: $\mathbf{AS} \xrightarrow{\mathbf{K}} \mathbf{BS}$, where $\mathbf{S} = [S_1 \cdots S_N]^\top$ represents the $N$-dimensional column vector of molecular species, $\mathbf{K} = [k_1 \cdots k_M]^\top$ the $M$-dimensional column vector of kinetic constants, and $\mathbf{A}, \mathbf{B} \in \mathbf{N}^{M \times N}$ the so-called stoichiometric matrices whose (non-negative) elements $[A]_{i,j}$ and $[B]_{i,j}$ are the stoichiometric coefficients $a_{ij}$ and $b_{ij}$ of the reactants and the products of all reactions, respectively. Given an arbitrary RBM and assuming the law of mass-action [67, 68], the system of coupled ODEs describing the variation in time of the species concentrations can be derived as follows:

$$\frac{d\mathbf{X}}{dt} = (\mathbf{B} - \mathbf{A})^T [\mathbf{K} \circ \mathbf{X^A}], \tag{2}$$

where $\mathbf{X}$ is the $N$-dimensional vector of concentration values (representing the state of the system) at time $t$, the symbol $\circ$ denotes the entry-by-entry matrix multiplication (Hadamard product), and $\mathbf{X^A}$ denotes the vector-matrix exponentiation form [67]. Note that each species $S_j$ at time $t$ is characterized by its concentration $X_j$, where $X_j \in \mathbf{R}^{\geq 0}$ for $j = 1, \ldots, N$. Formally, $\mathbf{X^A}$ is a $M$-dimensional vector whose $i$-th component is equal to $X_1^{Ai1} \cdots X_N^{AiN}$, for $i = 1, \ldots, M$. Note that, since we assume the law of mass-action, each ODE composing the system described in Eq 2 is a polynomial function with at least one monomial associated with a specific kinetic constant.

### Synthetic RBM generation

In order to investigate the computational performance of FiCoS, both the sets of symmetric and asymmetric RBMs of increasing size were randomly generated by exploiting a custom tool developed by our group, named SBGen [69], which allows for creating realistic RBMs whose dynamics resemble those of real biological networks. The synthetic RBMs satisfy the following characteristics:

- a log-uniform distribution in the interval $[10^{-4}, 1)$ was applied to sample the initial concentrations of the molecular species;

- a log-uniform distribution in the interval $[10^{-6}, 10]$ was used to sample the values of the kinetic constants;

- the stoichiometric matrix $\mathbf{A}$ was generated allowing only zero, first, and second-order reactions (i.e., at most two reactant molecules of the same or different species can appear in each

reaction). The rationale is that the probability of a reaction simultaneously involving more than two reactants is almost equal to zero. Anyhow, FiCoS can simulate RBMs with reactions of higher-orders;

- the stoichiometric matrix **B** is created allowing at most two product molecules for each reaction.

It is therefore clear that the matrices **A** and **B** are sparse. A log-uniform distribution (i.e., a uniform distribution in the logarithmic space) was used to sample both the initial conditions and the kinetic constants to capture the typical dispersion of both the concentrations and kinetic parameters, which span over multiple orders of magnitude [3, 70].

For each model, the initial value of the kinetic parameters was perturbed to generate different parameterizations (up to 2048). For each kinetic parameter $k_i$, with $i = 1, \ldots, M$, we applied the following perturbation:

$$k_i = \exp(\ln(k_i - 0.25 \cdot k_i) + (\ln(k_i + 0.25 \cdot k_i) - \ln(k_i - 0.25 \cdot k_i)) \cdot \text{rnd}), \qquad (3)$$

where rnd $\sim$ Uniform(0, 1) is a random number sampled from the uniform distribution in [0, 1].

## Implementation of FiCoS

In order to efficiently deal with both stiff and non-stiff systems of ODEs, FiCoS relies on two advanced integration methods belonging to the Runge-Kutta families: (*i*) the DOPRI explicit method [28–30] for solving non-stiff systems; (*ii*) the Radau IIA implicit method [31, 32] when the system under investigation is characterized by stiffness. In particular, we exploited DOPRI5 and RADAU5 methods, which are both Runge-Kutta methods of order 5. We also point out that our implementation was inspired by Blake Ashby's source code, who ported to C++ the original Fortran code developed by Hairer and Wanner [30, 31].

DOPRI is one of the most widespread explicit Runge-Kutta method, since it is a variable step-size algorithm and implements a stiffness detection approach [30]. Implicit Runge-Kutta methods are single-step approaches characterized by both a higher order of accuracy and more favorable numerical stability properties with respect to BDF algorithms [71]. Among the existing implicit Runge-Kutta methods, the Radau IIA family was proved to hold several useful properties: it is strongly *A*-stable and strongly *S*-stable, implying that it is both *A*-stable and *S*-stable, as well as stiffly accurate [72]. One of the main advantages of the *S*-stable methods is their ability to produce stable computations when solving highly stiff problems. In addition, when a fixed step-size is used, strongly *S*-stable methods are more accurate than the other methods. Several numerical tests showed that only *S*-stable methods can produce accurate simulations by using significantly large step-size for stiff problems [73]. Finally, RADAU5 is capable of arranging the step-size along with the desired error—controlled by means of $\varepsilon_a$ and $\varepsilon_r$ tolerances, which are user-defined parameters—and the detected level of stiffness.

We exploited a coarse-grained strategy to perform a large number of simulations in a parallel fashion, by distributing the simulations over the available GPU cores. Moreover, in order to accelerate the ODE resolutions of each simulation, we exploited a fine-grained strategy using the DP to distribute the integration algorithms on the GPU cores. When DP is used, each thread belonging to the grid called by the host (named parent grid) can launch a novel grid (named child grid) composed of several threads. All child grids that are executed by the thread of the parent grid can be synchronized, so that the parent threads can consume the output produced by the child threads without involving the CPU. Specifically, we leveraged the DP to distribute the calculations related to the Butcher tableau (see Section "GPU implementation" in

the S1 Text, for additional details) characterizing our integration methods. As a side note, FiCoS also exploits the cuBLAS library [74] for both the matrix decompositions and the linear system resolutions required by RADAU5.

Overall, FiCoS workflow consists in 5 different phases and 26 CUDA kernels, which can be summarized as follows:

- **Phase $P_1$**: it implements the generation of the data structures encoding the system of ODEs. These data structures are used to evaluate the ODEs during the resolution of the system.

- **Phase $P_2$**: for each simulation, the estimation of the dominant eigenvalue of the Jacobian matrix **J** is performed. Every simulation characterized by a dominant eigenvalue lower than 500 will be solved exploiting the DOPRI5 method. The other simulations will be tackled by means of the RADAU5 method. As a matter of fact, when **J** has a large spectral radius, the system of ODEs can be considered stiff [35].

- **Phase $P_3$**: it implements the DOPRI5 method. Note that if the DOPRI5 fails at solving some simulations, they are re-executed and solved relying on the RADAU5 method. It is worth noting that each thread, associated with a specific simulation, exploits the DP to call the kernels, developed to distribute the calculations required by DOPRI5, launching a novel grid of threads.

- **Phase $P_4$**: it implements the RADAU5 method. As in the case of the DOPRI5 method, each thread takes advantage of the DP to run the kernels.

- **Phase $P_5$**: for each simulation, the dynamics of the species are stored and written in the output files.

Notice that only phases $P_1$ and $P_5$ are executed on the CPU, while the others are entirely executed on the GPU.

## Supporting information

**S1 Text. It describes the design and implementation of FiCoS.** In addition, it shows the achieved results in more details.
(PDF)

**S2 Text. It explains the files required by FiCoS to perform the desired simulations.**
(PDF)

## Author Contributions

**Conceptualization:** Andrea Tangherloni, Marco S. Nobile, Paolo Cazzaniga.

**Data curation:** Andrea Tangherloni, Marco S. Nobile.

**Formal analysis:** Andrea Tangherloni, Marco S. Nobile, Paolo Cazzaniga, Giulia Capitoli, Simone Spolaor, Leonardo Rundo, Giancarlo Mauri, Daniela Besozzi.

**Funding acquisition:** Giancarlo Mauri, Daniela Besozzi.

**Investigation:** Andrea Tangherloni, Marco S. Nobile, Paolo Cazzaniga.

**Methodology:** Andrea Tangherloni, Giulia Capitoli, Daniela Besozzi.

**Project administration:** Daniela Besozzi.

**Resources:** Giancarlo Mauri, Daniela Besozzi.

**Software:** Andrea Tangherloni.

**Supervision:** Paolo Cazzaniga, Daniela Besozzi.

**Validation:** Andrea Tangherloni, Marco S. Nobile, Paolo Cazzaniga, Giulia Capitoli, Leonardo Rundo, Daniela Besozzi.

**Visualization:** Andrea Tangherloni, Marco S. Nobile, Paolo Cazzaniga, Leonardo Rundo.

**Writing – original draft:** Andrea Tangherloni, Marco S. Nobile, Paolo Cazzaniga.

**Writing – review & editing:** Andrea Tangherloni, Marco S. Nobile, Paolo Cazzaniga, Giulia Capitoli, Simone Spolaor, Leonardo Rundo, Giancarlo Mauri, Daniela Besozzi.

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
