## [Decision Letter · Decision Letter 0]

23 May 2021

Dear Dr Tangherloni,

Thank you very much for submitting your manuscript "FiCoS: a fine- and coarse-grained GPU-powered deterministic simulator for biochemical networks" for consideration at PLOS Computational Biology.

As with all papers reviewed by the journal, your manuscript was reviewed by members of the editorial board and by several independent reviewers. In light of the reviews (below this email), we would like to invite the resubmission of a significantly-revised version that takes into account the reviewers' comments.

Specifically, please make the source, along with the relevant tutorials and examples available as part of an open source repository, such as github and include the link in the manuscript. Otherwise the reviewers can not assess the software and provide an evaluation. This is an important part of software submissions.

We cannot make any decision about publication until we have seen the revised manuscript and your response to the reviewers' comments. Your revised manuscript is also likely to be sent to reviewers for further evaluation.

Sincerely,

Dina Schneidman

Software Editor

PLOS Computational Biology

Reviewer's Responses to Questions

**Comments to the Authors:**

Reviewer #1: I could not find any mention of where to get the software FiCoS, Without that I cannot review the paper properly. Even a search on google did not bring up any refernce to the software.

Reviewer #2: This paper presents a novel approach for conducting fine- and coarse-grain parallel simulations of ODE systems arising from biochemical reaction networks. The introduction provides a fairly comprehensive overview of previous efforts in this area and the body of the paper presents results of a series of comparisons between the new method and previous ones, including the widely-used standard single processor CPU-based implementations. Most of the comparisons use synthetic networks, but the final application is a case study on a large-scale existing model of autophagy, which demonstrates that the FiCoS approach is capable of performing a much larger number of ODE integrations given a fixed time budget. The method is described schematically in the Methods section at the end of the paper and in much greater detail in Supplemental material. Overall, this division of material seems appropriate. The paper is generally well written and easy to follow.

I have two main concerns about the paper. First, I believe this is intended to be an open source freely available tool (e.g., line 232 refers to the tool as "open source") but this is not made clear in the text and no links to the current implementation are provided. I feel that the significance of this work for the computational biology community would be greatly diminished if the tool and code were not available. I am sure that this point is easily addressed. The second concerns demonstration of the practical relevance of the tool. GPU's offer the potential to greatly increase the scale of computational analysis that can be performed on biological models, and the case study presented highlights how the method can be applied to a parameter sweep analysis, which is a commonly-performed task. However, I think it would strengthen the impact and interest of this paper to present demonstrations of one or two other common analyses, such as parameter sensitivity and parameter estimation, and to demonstrate how GPU-based simulations can be used in such practical applications to obtain superior performance. I would strongly encourage the authors to present their case studies in the form of Jupyter notebooks that would be available for the community to adapt for use in their own applications. I think a few examples like this could greatly increase the adoption of GPU use in the field compared to the current state, where I think most practitioners are not using this technology.

**Have the authors made all data and (if applicable) computational code underlying the findings in their manuscript fully available?**

Reviewer #1: None

Reviewer #2: **No: **

PLOS authors have the option to publish the peer review history of their article (what does this mean?). If published, this will include your full peer review and any attached files.

Reviewer #1: No

Reviewer #2: No
---

## [Decision Letter · Decision Letter 1]

12 Jul 2021

Dear Dr Tangherloni,

Thank you very much for submitting your manuscript "FiCoS: a fine- and coarse-grained GPU-powered deterministic simulator for biochemical networks" for consideration at PLOS Computational Biology. As with all papers reviewed by the journal, your manuscript was reviewed by members of the editorial board and by several independent reviewers. The reviewers appreciated the attention to an important topic. Based on the reviews, we are likely to accept this manuscript for publication, providing that you modify the manuscript according to the review recommendations.

Sincerely,

Dina Schneidman

Software Editor

PLOS Computational Biology

[LINK]

Reviewer's Responses to Questions

**Comments to the Authors:**

Reviewer #1: This is an interesting paper on speeding up mass-action based chemical models using GPUs. In fact this is the first paper I've seen where I am actually impressed with the performance. There are limitations, in particular the approach only deals with mass-action kinetics and as far as I can tell it cannot deal with nonlinear rate laws such as Hill of Michaelis-Menten like expressions. Do the authors have any plans to extend the applicably to more complex rate laws? A mention of this could be made in the discussion, eg what would be the issue to implement such capabilities?

The real bottleneck today in computational systems biology is parameter fitting where most of the time is spend in solving the ODEs. In our own work it can take 36 hours to fit a single model on a 8 core machine and we have many 1000s to fit. The second reviewer also mentioned this problem but it doesn't appear to have been addressed by the reviewers. The authors might not be able to solve this problem at the moment but there should be some comment on this is the discussion section and whether a GPU approach could solve this.

In terms of comparison it might have been better to compare against CVODE rather than LSODA. CVODE is a much more modern integrator, LSODA is now over 25 years old is unchanged during that time.

No mention of SBML is made at all in the paper or supplement. Most models are now stored in SBML (see biomodels), how are users to exploit the new GPU code given that models are in SBML? As a result no comparisons were made with existing simulators such as COAPSI, VCell, roadrunner etc.

I was very happy to see that code is now open source. Note that the use of GPL-3 will likely significantly restrict it use by the community., perhaps a more liberal open license would be better? Of course the authors have the final say in the type of license they use.

Finally, the supplement mentioned the hardware that was used, "All tests were performed on a workstation equipped with a CPU Intel Core i7-2600 CPU (clock 3.4 GHz) and 8 GB of RAM, running on Ubuntu 16.04 LTS. The GPU used in the tests was a Nvidia GeForce GTX Titan X (3072 cores, clock 1.075 GHz, RAM 12 GB), CUDA toolkit version 8 (driver 387.26)."

I would move this to the methods section in the main text because if someone want to use this code, they need to know what hardware to purchase and the method section the best place for this. For example this reviewer is interested in trying out the code.

**Have the authors made all data and (if applicable) computational code underlying the findings in their manuscript fully available?**

Reviewer #1: Yes

PLOS authors have the option to publish the peer review history of their article (what does this mean?). If published, this will include your full peer review and any attached files.

Reviewer #1: No

Figure Files:

Data Requirements:

Reproducibility:

References:

---

## [Decision Letter · Decision Letter 2]

28 Aug 2021

Dear Dr Tangherloni,

We are pleased to inform you that your manuscript 'FiCoS: a fine-grained and coarse-grained GPU-powered deterministic simulator for biochemical networks' has been provisionally accepted for publication in PLOS Computational Biology.

Best regards,

Dina Schneidman-Duhovny

Software Editor

PLOS Computational Biology

Dina Schneidman-Duhovny

Software Editor

PLOS Computational Biology

Reviewer's Responses to Questions

**Comments to the Authors:**

Reviewer #1: Many thanks to the authors, all my comments have been resovled. This was an interesting paper.

**Have the authors made all data and (if applicable) computational code underlying the findings in their manuscript fully available?**

Reviewer #1: Yes

PLOS authors have the option to publish the peer review history of their article (what does this mean?). If published, this will include your full peer review and any attached files.

Reviewer #1: No

---

## [Editor Report · Acceptance letter]

6 Sep 2021

PCOMPBIOL-D-21-00064R2 

FiCoS: a fine-grained and coarse-grained GPU-powered deterministic simulator for biochemical networks

Dear Dr Tangherloni,

I am pleased to inform you that your manuscript has been formally accepted for publication in PLOS Computational Biology. Your manuscript is now with our production department and you will be notified of the publication date in due course.

With kind regards,

Andrea Szabo
